# Smart Brace for Static and Dynamic Knee Laxity Measurement

**DOI:** 10.3390/s22155815

**Published:** 2022-08-04

**Authors:** Paolo Bellitti, Michela Borghetti, Nicola Francesco Lopomo, Emilio Sardini, Mauro Serpelloni

**Affiliations:** Department of Information Engineering, University of Brescia, Via Branze 38, 25123 Brescia, Italy

**Keywords:** IMU, stretchable strain sensors, wearable, knee laxity, ACL, knee biomechanics

## Abstract

Every year in Europe more than 500 thousand injuries that involve the anterior cruciate ligament (ACL) are diagnosed. The ACL is one of the main restraints within the human knee, focused on stabilizing the joint and controlling the relative movement between the tibia and femur under mechanical stress (i.e., laxity). Ligament laxity measurement is clinically valuable for diagnosing ACL injury and comparing possible outcomes of surgical procedures. In general, knee laxity assessment is manually performed and provides information to clinicians which is mainly subjective. Only recently quantitative assessment of knee laxity through instrumental approaches has been introduced and become a fundamental asset in clinical practice. However, the current solutions provide only partial information about either static or dynamic laxity. To support a multiparametric approach using a single device, an innovative smart knee brace for knee laxity evaluation was developed. Equipped with stretchable strain sensors and inertial measurement units (IMUs), the wearable system was designed to provide quantitative information concerning the drawer, Lachman, and pivot shift tests. We specifically characterized IMUs by using a reference sensor. Applying the Bland–Altman method, the limit of agreement was found to be less than 0.06 m/s^2^ for the accelerometer, 0.06 rad/s for the gyroscope and 0.08 μT for the magnetometer. By using an appropriate characterizing setup, the average gauge factor of the three strain sensors was 2.169. Finally, we realized a pilot study to compare the outcomes with a marker-based optoelectronic stereophotogrammetric system to verify the validity of the designed system. The preliminary findings for the capability of the system to discriminate possible ACL lesions are encouraging; in fact, the smart brace could be an effective support for an objective and quantitative diagnosis of ACL tear by supporting the simultaneous assessment of both rotational and translational laxity. To obtain reliable information about the real effectiveness of the system, further clinical validation is necessary.

## 1. Introduction

Within a physiologically healthy human knee joint, the relative movements between the tibia and the femur are passively constrained by the presence of different ligamentous structures, which mainly limit the anteroposterior translation, internal-external and varus-valgus rotations. However, small excursions (i.e., laxity) are still possible depending also on the conditions of the ligaments themselves; small ranges of translations and rotations are usually considered physiological, whereas “big” excursions are in general associated with ligaments that are no longer able to function as a constraint. This increase in joint laxity could be associated with pathological conditions in which one or more ligaments are “loose”. In fact, severe traumas involving the knee joint can result in ligament lesions, which lead to an increase in specific joint laxity. In this context, anterior cruciate ligament (ACL) tears represent one of the most common injuries affecting the knee, with a yearly incidence estimated to be between 30 and 78 per 100,000 people [1]. Usually, these lesions are surgically treated, leading to about 80,000 to 100,000 ACL reconstructions performed annually in the United States [2].

In this context, the assessment of knee laxity represents one of the most important tasks to be performed for ACL tear diagnosis and treatment selection as the degree of laxity is a key factor to take into account when selecting the appropriate option for reconstruction [3,4]. Concerning this diagnosis, it is fundamental to identify the performance of any specific clinical test, in terms of sensitivity and specificity, which are the ability to correctly identify those subjects with the lesion, and the capability to correctly recognize those subjects without the tear, respectively [5]. Several clinical tests have been adopted to evaluate ACL status through the assessment of knee joint laxity; in general, three are the most used for their diagnostic performance, namely:

Lachman test: This is the most sensitive laxity test used in clinical evaluation and has long been considered the gold standard in detecting an ACL injury [6]; this test well highlights the ability of ACL to statically constrain the anterior displacement of the tibia with respect to the femur.Anterior drawer test: This is a widely used assessment that—similarly to the Lachman test—tries to underline the ACL function in the sagittal plane; however, it represents a poor diagnostic indicator of ACL tear, especially when considering acute settings [7].Pivot-shift test (PST): This is a complex test able to highlight the dynamic laxity of the knee joint and represents the most specific test for ACL lesion, thus being the benchmark for ACL injury assessment [6,8].

Although PST represents the standard and the most used test to evaluate the dynamic laxity, it has been demonstrated that the outcomes are dependent on the examiner since the stress on the joint is manually applied, and the feedback can be highly variable because of the “feeling” perceived by the tester and his/her experience [9]. Indeed, the PST is a complex maneuver, since—aiming at highlighting rotatory knee laxity—it requires a combination of axial load and valgus stress applied during the knee flexion, thus being highly sensitive to reproducibility errors.

Therefore, the need for a quantitative evaluation system has risen during the last decade. In the literature, there are essentially four types of devices to quantitatively measure the pivot shift test: electromagnetic tracking devices [10,11], image-based techniques [12,13], computer-assisted surgery (CAS) including navigation systems [14,15,16], and inertial sensors [17,18,19]. All these solutions present several advantages but also drawbacks. Electromagnetic tracking devices, through the application of a varying magnetic field to establish the position and orientation of the sensors, can provide accurate and reliable measurements and can be used in a non-invasive way; however, this technology is highly subject to interference due to the presence of metallic objects within the tracking volume and, because of their non-invasive nature, are affected by soft tissue artifacts [20,21]. Motion tracking solutions based on the use of cameras and image analysis represent a novel approach for dynamic knee laxity evaluation. Even though these systems can be considered cheap and non-invasive, at present, they reported—besides the problems related to soft tissue artifacts—low sensitivity and poor correlation with actual bony movements, which is indeed one of the main challenges in ACL tear diagnosis [22]. Computer-assisted surgery (CAS), including navigation systems, represented the first and one of the most important techniques used for measuring both static and dynamic laxity. These solutions exploiting the possibility to securely fix the tracker on the bones during the surgery, are extremely reliable and repeatable since they are not inherently subject to soft tissue artifacts and muscular guarding [4]. However, they are invasive, highly expensive and cannot be used in inpatient or outpatient settings. Finally, inertial measurement units (typically integrating accelerometers, gyroscopes, and magnetometers) have been proved to be simple and cheap solutions that can be used in the quantitative assessment of PST [8,22]; indeed, inertial sensors are small and noninvasive and they can be connected (even wirelessly) to a common laptop or mobile/tablet for data elaboration [17,18,19]. Using this technique, one sensor is usually attached to the tibia to provide information about the acceleration of the segment during the maneuver; the results obtained with this kind of system proved a strong correlation between the clinical grade of the pivot shift test and a decrease in femoral acceleration during tibial reduction [18]. However, further validations and reliability tests are still needed for this technology, above all, when the relative rotation between the femur and tibia is required; furthermore, when using this approach, soft tissue artifacts may cause errors in the estimation of relative movements [22].

Even though a great variety of systems have been developed in the recent past to quantify the knee laxity—including both static and dynamic tests—each method presents advantages and disadvantages that must be carefully evaluated when considering a specific solution. Focusing on dynamic laxity, it is fundamental to reduce the effects of soft tissues and minimizing the variability due to the examiner’s experience; in this perspective, a possible solution is given by mechanized PST. In addition, future studies still have to gather data and focus on the reliability, validity, and comparisons of kinematic outcomes with respect to clinical grading. From this perspective, the possibility of obtaining reliable information concerning both the static and dynamic behavior of the joint under specific stresses is indeed of extreme interest to clinicians.

In this work, a “smart” brace is proposed as an integrated and affordable solution able to provide quantitative information about knee laxity, thus supporting the clinical assessment in both inpatient and outpatient settings. The proposed smart brace is equipped with two inertial measurement units (IMUs) for knee joint angle measurements and three strain sensors for quantifying displacements associated to both translational and rotational laxities of the knee joint. The smart brace is proposed to be used during the administration of several clinical tests—i.e., the Lachman test, Anterior drawer test and PST—in support of an objective diagnosis of ACL tear. Following the Introduction, Section 2 describes the smart brace design, and Section 3 reports the experimental results on the characterization of the stretch sensors and the validation of the IMUs. Section 4 examines the overall validation protocol of the integrated smart brace, and Section 5 reports and discusses the preliminary results obtained from a first proof-of-concept study.

## 2. Smart Brace Design

To obtain information about both static and dynamic laxities and taking into account the degrees of freedom of the knee joint and the stressing tests, the “smart” brace was based on a knee brace equipped with three stretch sensors and two wearable IMUs, as shown in Figure 1. Starting from the analysis of literature [23,24], the insertions of the stretch sensors and the positions of the IMUs was defined according to the hypotheses of acquiring both rotational and displacement information during the Lachman, drawer, and pivot shift tests. The block diagram of the smart brace and the data collection system is shown in Figure 2.

The stretch sensors (Images SI Inc., New York, NY, USA) are conductive rubber cords of 1.7 mm in diameter and 100 mm long, whose electrical resistance changes when they are stretched [25]. When relaxed, the sensors have a nominal resistance of 1400 Ω. The stretch sensors were securely sewn to the brace and were positioned in order to recognize the key components of knee laxity with the minimum number of strain sensors and according to a recent study [23]. Referring to Figure 3, Stretch Sensor 1 addresses medial-lateral translation, Stretch Sensor 2 the internal-external rotation, and Stretch Sensor 3 the anterior and posterior translation. Stretch Sensor 2 is the one attached to the femur epicondyle that—we hypothesized—is able to capture components of rotational laxity.

The SparkFun 9DoF Razor IMU M0 (“IMU device”, SparkFun, Niwot, CO, USA) was selected as the IMU device with the following embeds:-The MPU-9250 system, a 9-axis MEMS sensor, which includes a 3-axis accelerometer, a 3-axis gyroscope, and a 3-axis magnetometer to detect the movement of the femur/tibia segment. The full scale was set at 2 g for the accelerometer and 2000 dps for the gyroscope. The cut-off frequency of the digital low pass filter was set to 5 Hz with a sampling frequency of 40 Hz. The Digital Motion Processor (DMP) inside the MPU-9250 merges the acceleration and rotation values provided by the sensors to minimize the errors that affect each sensor and calculates the device orientation in terms of quaternions, from which the rotation information can be extracted. The DMP was configured in order to have a quaternion every 25 ms.-A 32-bit ARM Cortex-M0+ processor with 256 KB flash, 32 KB SRAM, and an operating speed of up to 48 MHz. It was used to (1) read the output of the stretch sensors, (2) collect and elaborate the measurements of the inertial sensors, and (3) send all data to an external wireless module (XBee Module, XB24CAWIT-001, DIGI, Hopkins, MN, USA) via the I^2^C header.-An LiPo charger and a power switch.

In order to minimize the effects of soft tissue artifacts, the tibial IMU was placed in the proximal anterolateral part of the tibia, between the anterior tuberosity and the Gerdy’s tubercle, while the femoral IMU was placed in the distal lateral part of the femur, near the epicondyle. In order to reduce the encumbrance of the cables, two different boards were developed: the “femoral” unit containing one IMU and the front-end for the Stretch Sensor 1, and the “tibial” unit containing the second IMU and the front-end for Stretch Sensor 2 and Stretch Sensor 3. Each unit includes a lithium-ion battery with nominal 3.7 V at 1000 mAh (PRT-13813, SparkFun, Niwot, CO, USA) for powering the portable IMU device, and a wireless XBee transmitter connected to the IMU device for communicating with a software application. XBee modules communicate using the Zigbee protocol and guarantee low-power consumption and easier management of the mesh network.

As mentioned before, the measurement of each stretch is obtained by conditioning the sensor output with a front-end (labeled as “Passive Network” in Figure 2). In order to simplify the analog-to-digital conversion of the sensor output in terms of size and power consumption [26,27], a direct microcontroller interface (DIC) was selected. The microcontroller excites with a voltage *V*_0_ the sensor in series with a fixed capacitor of capacitance *C* and then measures the discharge time *T_m_* from the maximum value and a predefined threshold *V_th_* when the excitation signal is set at 0 V. With this method, the resistance *R* of the sensor can be calculated with Equation (1):(1)$$R=\frac{T_{m}}{C\cdot \ln\left(\frac{V_{0}}{V_{th\;}}\right)}$$

Each of these sensors is connected on one side to a digital pin used as an output (*V*_0_ = 3.3 V), and on the other side to the capacitor. The capacitor is connected to the ground and to one of the analog inputs of the microcontroller ADC (internally connected to a digital timer). The current absorbed by the analog input can be considered negligible since the analog input has a very high input resistance. The threshold *V_th_* was selected to be equal to 36.8% of *V*_0_ in order to simplify Equation (1) to Equation (2).
(2)$$R=\frac{T_{m}}{C}$$

The measurements of both units were sent to an acquisition system composed of an Xbee module that worked as a coordinator connected to a computer and custom-made software (written in LabVIEW2017, National Instruments, Austin, TX, USA) to display, synchronize, and store the measurements of each board.

## 3. Stretch Sensor and IMU Characterization

It was experimentally verified that no packet loss occurred over one hour, even when both units sent data to the coordinator at the same time.

The current consumption of each unit was experimentally measured during the normal operation, which includes the reading of the sensors (IMUs and stretch sensors), the elaboration and the transmission of the data to the coordinator. A benchtop multimeter (HP34401A, Agilent, Santa Clara, CA, USA) was used for this test. The average current required for measuring and transmitting the data to the coordinator was 45 mA. The power consumption is in accordance with the specification of the IMU unit and it was obtained when the resistance of the three stretch sensors was at its minimum value (i.e., 1400 Ω). Considering the battery capacity (1000 mAh) and according to the experimental tests, the overall power consumption guarantees 20 h of operation, which is far longer than a single clinical session time for the laxity evaluation and can cover a daily outpatient routine.

The gauge factor (GF) was found for the three stretch sensors (Stretch Sensor 1, Stretch Sensor 2, and Stretch Sensor 3), defined as in Equation (3):(3)$$GF=\frac{\frac{R-R_{0}}{R_{0}}}{\frac{L-L_{0}}{L_{0}}}=\frac{\frac{\Delta R}{R_{0}}}{\varepsilon }$$
where $L_{0}$ and $R_{0}$ are the length of the sensor (100 mm) and the resistance before the elongation (i.e., no strain is applied), respectively, and L and R are the length and the resistance when elongation is applied. The sensing properties of this type of sensor were extensively investigated in a previous work [25] through four different tests (i.e., strain ramp test, cyclic loading-unloading test, relaxation test, and temperature test). The experimental setup for GF estimation was the same as reported in a previous work [25]. As highlighted in [25], the ends of the sensor were firmly clamped in the two grips of the testing machine. One grip of this testing machine was motionless, and one was sliding and operated by a linear motor actuator. The motor position P was provided by an encoder mounted on the motor and connected via USB to a computer for the data acquisition. Starting from the position $P_{0}$ corresponding to unstretched status, the elongation of the sensors was changed from 0 mm to 100 mm at 0.15 mm/s. Considering an initial length of the unstrained sensor equal to 100 mm, the strain (ε) changed from 0% to 100% and it was calculated according to Equation (4):(4)$$\varepsilon =\frac{L-L_{0}}{L_{0}}=\frac{P-P_{0}}{P_{0}}$$

The resistance acquisition through a benchtop multimeter (HP34401A, Agilent, Santa Clara, CA, USA) and the measurement of the position of the motor are synchronized with a LabVIEW program. According to the results shown in Figure 4, the GF was similar for all the sensors (ranging from 2.163 to 2.175). This finding is also confirmed in Figure 5, where the measured data of the three sensors were averaged (square markers), while the maximum and the minimum values are represented by the error bars. The data can be fitted by a line with a slope of 2.169, which represents the GF. The deviation from the fitted line is less than 4.2% even for strains greater than 70%.

The stretch sensor is affected by relaxion effect under stress. Indeed, when the stretch sensor is elongated and then released to the initial condition, the time required to recover the initial value ($R_{0}$) is estimated at two minutes, but after 20 s from the release the resistance is only 5% more than the initial value ($R_{0}$). Considering the final application, where the clinician performs a rapid maneuver, the relaxation effect is negligible. The stretch is also affected by the overshoot when it is elongated (10% of the final value), as reported in the literature, but it is negligible in the final application since the resistance measurement is obtained through the DIC architecture.

The IMUs were calibrated to determine the offset, the scale factor, and the misalignment matrix with respect to the device body axes.

The accelerometer was calibrated according to the static method described in [28]. The platform was rotated through 36 positions and kept in each position for 20 s. Firstly, the sensor was placed flat with the IMU z-axis aligned to the gravity vector and the IMU x-axis parallel to the ground. The platform was rotated clockwise around the y-axis of 30 degrees, from 0 degrees to 360 degrees, and the data related to the accelerometer sensor were stored. Then the platform was placed with the IMU x-axis aligned to the gravity vector and the IMU y-axis parallel to the ground, and rotated 30 degrees clockwise around the z-axis of 30 degrees until the sensor was rotated 360 degrees. Finally, the platform was placed with the IMU y-axis aligned to the gravity vector and the IMU z-axis parallel to the ground, and then rotated 30 degrees clockwise around the x-axis of 30 degrees until the sensor was rotated 360 degrees. The measurements and the expected outputs were placed in the two matrices and the offset, the scale factor, and the misalignment matrix were calculated according to the equation in [28].

The gyroscope offsets were calibrated automatically by the software when the sensor was in the static condition (no movements and placed on the table). The scale factors were determined by rotating the sensor around each device body axis.

The magnetometer was calibrated by using the least square fitting ellipsoid method. In this way, the hard-iron, soft-iron, and scale factors were compensated.

After the calibration, the calibration parameters were updated in the firmware. The measurements of the IMUs were validated by using a more accurate commercial system (MTw Awinda, XSens, Enschede, The Netherlands). The IMU to be calibrated and the MTw sensor were placed on the same platform by aligning the axes of both accelerometers/gyroscopes. For the synchronization of sensor data, a synchronization gesture was performed before the tests, and then the platform was moved and rotated randomly in every direction.

The measurements from the IMU tibial sensor and the MTw sensor are shown in Figure 6, Figure 7 and Figure 8. The Bland–Altman plots are used to assess the agreement between these two devices, comparing the measurements of the accelerometers (Figure 6), gyroscopes (Figure 7), and the magnetometers (Figure 8) for each axis. The bias is always not consistent due to the calibration of the sensors, and no relevant relationships between the discrepancies and the true value are detected in the plots. Few outliers were found and the limits of agreement, defined as the mean difference plus (and minus) 1.96 times the standard deviation of the differences indicate good reliability of the IMU sensors.

## 4. Validation Protocol

A marker-based stereophotogrammetric system (Smart-Dx 400, BTS Bioengineering, Milan, Italy) usually used for human motion analysis, was adopted to acquire kinematic data for the validation of the “smart” brace. The system consisted of eight digital cameras equipped with infrared illuminators to detect and reconstruct the trajectory in three-dimensional space of a set of passive spherical markers (11 mm diameter) attached to the subject’s body through rigid trackers. The camera resolution is 0.3 Mpixel, and the accuracy in identifying the 3D position of a single marker was—after proper calibration—lower than 0.2 mm on a 1.5 × 1.5 × 1.5 m^3^ volume. The sampling rate was set at 100 Hz.

The data acquired with the digital cameras (i.e., the trajectories of the markers attached to the rigid trackers [29,30]) were reported with respect to the global reference frame of the tracking system and they needed to be transferred to anatomical coordinates in order to be comparable with the IMUs data. In order to do this, after system calibration, a registration phase was performed through the acquisition of several anatomical landmarks with a dedicated tracked wand [30]. The identified landmarks were specifically:hip center (identified through pivoting);femoral epicondyles;tibial plateau extremities;tibial malleoli.

The anatomical reference system adopted has been studied in [31,32]. The femoral anatomical reference system was defined by setting the origin in the middle point between the epicondyles, the z-axis along the femur, and pointing out the hip center identified through a least-square optimization algorithm. The x-axis was set as the transepicondylar line normalized with respect to the z-axis, and the y-axis as the cross product between the z-axis and x-axis. Similarly, the tibial anatomical reference system was defined by identifying the origin as the midpoint between the two plateaus, the z-axis as the line joining the origin with the midpoint between the two malleoli, the x-axis as the line joining the tibial plateau extremities, and the y-axis as the cross product between z-axis and x-axis. The x-axis was then normalized by considering the z-axis as the most reliable. The six degrees of freedom (DoF) were the computed by considering the relative motion between the tibial reference frame and the femoral one via the Grood and Suntay algorithm, so as to obtain instantaneous rotations and displacements in the anatomical planes.

Similarly, an anatomical registration phase was also performed for the IMU setting in order to obtain sensors-anatomy alignment [33]. To obtain this information, the subject was made to stand still—to obtain gravitational acceleration as an indication of the vertical direction (i.e., z-axis for the thigh and shank segments)—and then a passive flexion-extension movement was performed on the knee in order to identify the knee flexion-extension axis (i.e., the x-axis for thigh and shank segments); during this last prototypical movement, the lower limb was moved with a series of flexion-extension involving the knee from 0° to about 90° without introducing intra-extra or varus-valgus rotations. Information about this passive movement was also considered in the analysis reported in Section 5.

After the registration, several tests were performed in order to validate the designed system with the help of an orthopedic surgeon and four volunteers self-reporting instability of the knee and probable torn ACL. A preliminary analysis of the risks associated with the experimental session was performed. All the subjects were clearly informed about the tests they were involved in and gave their explicit consent to participate in the validation. Three diagnostic tests were performed on each knee:The Lachman test is the most sensitive laxity test used in the clinical assessment of ACL injury [6]. This test is usually performed with the subject in a supine position with the knee maintained at about 15–20 degrees of flexion. The examiner stresses the knee joint in anterior direction by grasping the proximal part of the tibia and then pulling it anteriorly. This test is considered clinically positive if there is a “soft end” feel perceived by the examiner and/or there is a displacement of more than 2 mm compared to the contralateral limb [34].The drawer test is a test widely used in clinical practice, but it is a poor diagnostic indicator of ACL ruptures, especially in the acute setting [7]. The test is usually performed with the subject in supine position while maintaining the knee bent at approximately 90 degrees of flexion. The examiner, while firmly grabbing the proximal part of the tibia, stresses the joint by applying a force in the anterior (or posterior) direction. This test is considered clinically significant if there is a “soft end” feel or an excessive displacement (more than 5–6 mm compared to the contralateral limb) in the anterior (or posterior) direction.The pivot shift test (PST) is the most specific test and it represents the benchmark for ACL injury assessment [6,8], since the maneuver is able to test the dynamic laxity associated with ACL insufficiency. The main drawback of this test is the lack of standardization in the execution of the test itself, and also in the lack of objective grading [16]. The test is usually performed with the subject in a supine position, with the knee fully extended and the hip maintained flexed to about 30 degrees. The examiner, holding the heel, introduces a torque that leads the knee in internal rotation, and slowly flexes the knee while putting a moderate valgus moment on the proximal tibia. This test is considered clinically positive if the lateral tibial plateau subluxes anteriorly at about 30 degrees of knee flexion.

In our setting, each maneuver was repeated three times on both the healthy and injured knees of four subjects.

For each test carried out, correlation indices were calculated between the angles obtained by using the optoelectronic system (*opto*) with those estimated using the Madgwick algorithm applied to the IMUs data [35]. The results presented in Section 5 are the average results of all the maneuvers of the same type (for all the four subjects and both knees); the value of the initial angle was also subtracted from each signal to reduce the effects of non-alignment between the two systems adopted. Since there is a wide inter-subject variability in joint laxity due to specific anatomo-functional features, laxity assessment should be performed on both the knees in order to highlight any difference that is related to the ACL injury itself.

For brevity, the results from the optical system and with the inertial sensors will be indicated with the label *opto* and *imu* respectively. FE refers to the knee flexion and extension, AA refers to the adduction and abduction, and IE refers to the internal and external rotation

All data collected in the acquisition phase were post-processed through a custom mathematical toolbox developed in MATLAB (MathWorks^®^).

## 5. Results and Discussion

Focusing on the main principles on which the proposed device is based, we report the potential to obtain different kinds of information related to both rotational and translational components of the laxity and, then, provide evidence about the added value given by each sensor during the realization of different clinical tests.

From this perspective, Figure 9 provides an example of the data retrieved from the smart brace during the execution of pivot shift maneuver for the Stretch Sensor 2 displacement and IE angular displacement. Figure 9a refers to the pivot shift maneuver on a healthy knee and Figure 9b refers to an injured ACL.

Indeed, here we can recognize the main characteristics of the smart brace, which is able simultaneously to provide complementary information about the relative rotations (estimated via IMUs) and translation of the lateral compartment (estimated by using the Stretch Sensor 2), thus supporting better the diagnosis of ACL lesions.

Focusing then on the information available during different clinical tests, Figure 10 shows the calculated flexion and extension displacement (FE) of the knee obtained by IMUs (blue line) and the optoelectronic system (red line) during the passive flexion and extension test. The y-axis is scaled in order to better appreciate that the relative angular displacements over the range are very similar for both systems. The error on the scaled values, therefore, represents a relationship index that weights the variation of the angle over time more compared with the point-to-point error that weights the disparities in amplitude.

Table 1 summarizes the most relevant indices obtained for the neutral flexion-extension, pivot shift, Lachman, and anterior drawer tests. *Opto* span and *imu* span are the differences between the maximum and minimum angle of both systems. *E_max* is the absolute angle difference between the maximum values of *imu* and *opto*. *E_point* is the absolute average difference between the *imu* and *opto* signal, and *E_scal* is the relative *E_point* with respect to the span assumed during the test. For example, *E_scal* for FE of the knee is obtained by using Equations (5)–(7):(5)$$FE_{opto\;\%}=\frac{{\bar{FE}}_{opto}-min(FE_{opto})}{max(FE_{opto})-min(FE_{opto})}=\frac{{\bar{FE}}_{opto}-min(FE_{opto})}{Span\_opto}$$
(6)$$FE_{imu\;\%}=\frac{{\bar{FE}}_{imu}-min(FE_{imu})}{max(FE_{imu})-min(FE_{imu})}=\frac{{\bar{FE}}_{opto}-min(FE_{imu})}{Span\_imu}$$
(7)$$E\_scal_{imu\;\%}=\left|FE_{opto\;\%}-FE_{imu\;\%}\right|\;$$
where $FE_{opto}$ and $FE_{imu}$ are the scaled FE calculated from the *opto* system and *imu* system respectively, and ${\bar{FE}}_{opto}$ and ${\bar{FE}}_{imu}$ are their average values.

In the passive flexion-extension test, FE is the most relevant angle that varies during the test. According to the results in Table 1, in this test *Span_imu* is less *Span_opto* but *E_scal* is good. In the pivot shift test, similar results are obtained. In the Lachman test, all angles reached a span of less than 14°, and the errors range from 15% to 16% for FE and AA. It should be remembered that during the test the solicitation imposed by the doctor is only that relating to the AP. Similar results are obtained in the Drawer test, but here *E_scal* increases for all the angles, probably due to extremely low angular excursions. The estimation error of IE, which is consistent in all the indices, negatively affects the validity of the data obtained. Figure 11 shows the analysis of the angular displacement of IE during the pivot shift test in which it is possible to note the criticalities of the results.

Since for the drawer and Lachman tests the most stressed degree of freedom is the anteroposterior translation (AP), the analysis is focused on the stretch sensor applied to the anterior part of the knee (Stretch Sensor 3). The sensor signal obtained from each subject, referred to the initial instant, was compared to the maximum values (Figure 12). Overall, the tests confirm that the greatest elongation always occurs in the injured joint.

The Lachman test stresses the AP more. The results of Lachman test (Figure 13) show that the clinician’s diagnosis is respected in 75% of cases; the abnormal variation of Stretch Sensor 3 for Subject 3 could be attributable to involuntary traction of the brace, its incorrect positioning, or an active muscular contraction performed by the subject.

The pivot shift is the most complex test among the three evaluated because it stresses the joint in multiple ways. For this reason, the study focused on the point where the maneuver determines the anterior subluxation of the external tibial plateau with respect to the femur, and the internal rotation of the injured joint should be higher. During the test, the high excursion of the FE causes a proportional increase in Stretch Sensor 3, and secondarily, in Stretch Sensor 2, which is linked to internal rotation. The results obtained (Figure 14) show that in 75% of cases, the injured knee is wider than the healthy one. It was decided to relate the maximum value of the signal with that obtained from the flexion-extension tests conducted at the beginning of the protocol due to the complication arising from the different positioning of the brace from person to person; in fact, the tension of the elastic links of the support greatly affects the sensitivity of the transducer. In this way, a partial compensation of the amplitude variation phenomena due to the different positioning was obtained.

According to the obtained results, Stretch Sensor 1 seems to not provide significant information to highlight the effect of the ACL injury in both translational and rotational laxity, whereas the combination of the information provided by all the other sensors is more sensitive to ACL tear. It is worth noting that the detection of an ACL lesion is possible only by comparing the behavior of both the knees during the laxity tests due to the inter-subject variability, which is related to subjective anatomy, local morphotypes, muscular and soft tissue characteristics, training and other factors.

## 6. Conclusions

In this work, a new compact, easy-to-use, and cheap system for quantifying dynamic knee laxity has been designed, realized, and tested. This device could potentially fill the needs of orthopedic clinicians who aim to obtain quantitative information—even in inpatient and outpatient settings—and support them in the diagnosis of ACL tear. It can increase the overall sensitivity and specificity of the most used clinical tests, including the pivot shift.

In addition to the electronic design of the conditioning circuit and the mechanical challenges concerning the integration of sensing components on and within the knee brace, completely new software was coded both for the interface (LabVIEW, National Instruments) and for the microcontroller in the Arduino environment. The overall system was specifically designed to address the need for low power characteristics and can operate for about 20 h with a common 1000 mAh battery.

From a clinical perspective, the smart brace was functionally tested in a preliminary setting and the sensors were characterized. The system was tested on four subjects and the results were compared with an optoelectronic system, which indeed represents the actual state-of-the-art for benchmarking. The best results were obtained from the pivot shift test, as there is almost a perfect correlation between the optoelectronic and the stretch sensor when considering AP translation, and between the optoelectronic and IMUs when considering IE rotation. In the peak-to-peak analysis of the strains, it can be noted that there is generally a larger strain on the injured right knee which is consistent because of the increased laxity that occurs after an ACL tear. While the performance of the device is satisfactory, there is space for improvement considering the clinical setting. In particular, reliability in the positioning of the brace could be improved as well as the integration of the stretch sensors; furthermore, solutions including e-textiles should be considered for future development. In addition, a wider session of tests in a clinical trial are necessary to obtain a correct validation. In the near future, the system could be improved from both the design and technology perspective by including, for instance, an embedded algorithm that could synthetize data by integrating the sensors’ output and allow a user-friendly data presentation based on suitable thresholds and indicators. Furthermore, the device could be optimized by using a minimum number of sensors (above all considering the stretch sensors), which can be considered significant in the assessment of knee laxity, thus simplifying the complexity of the device itself and the interpretation of the measurements.

## Figures and Tables

**Figure 1 sensors-22-05815-f001:**
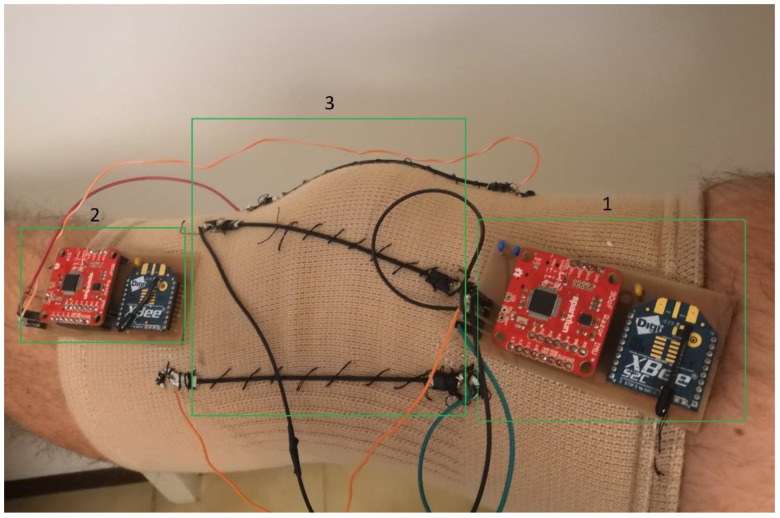
Smart left knee brace for dynamic knee laxity measurements. (1) Femoral unit (IMU, microcontroller, transceiver, and front-end for stretch sensors); (2) Tibial unit (IMU, microcontroller, transceiver, and front-end for stretch sensors), (3) three stretch sensors.

**Figure 2 sensors-22-05815-f002:**
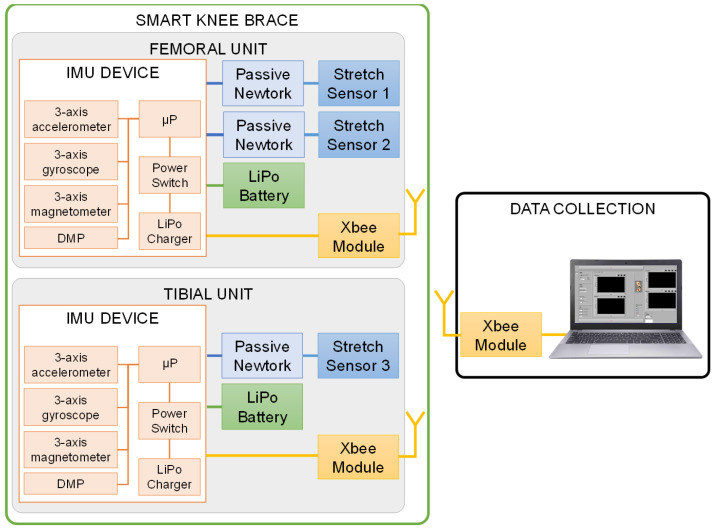
Block diagram of the smart knee brace and the data collection system.

**Figure 3 sensors-22-05815-f003:**
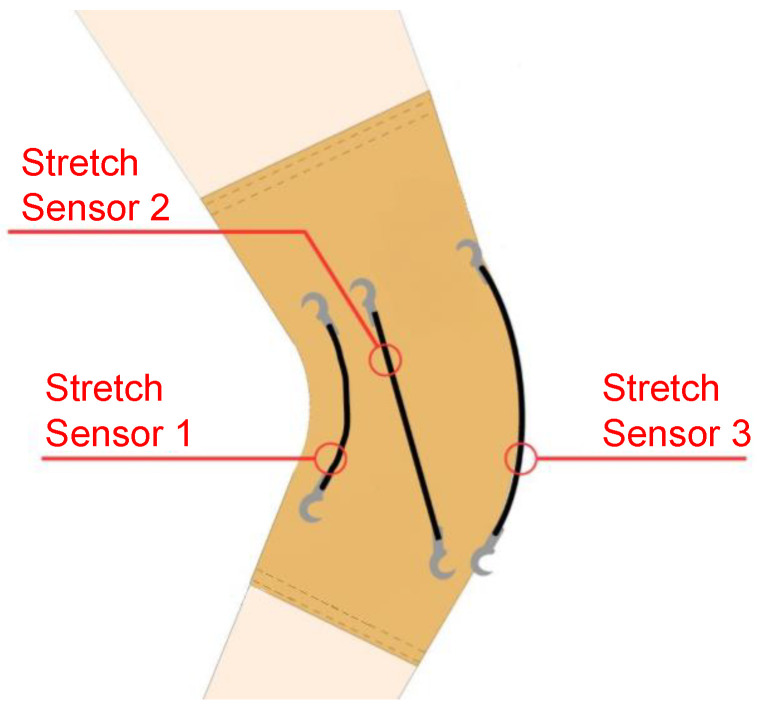
Position on the knee of the sensors evaluating knee laxity.

**Figure 4 sensors-22-05815-f004:**
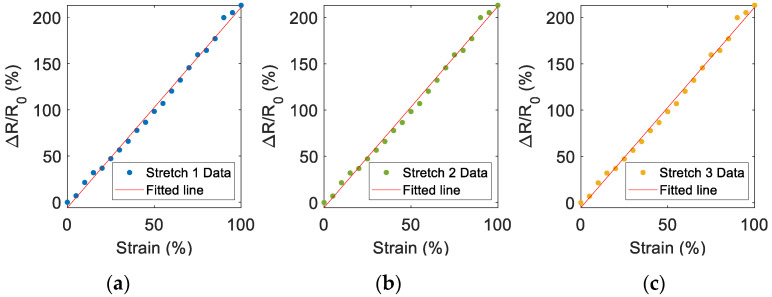
Response of (**a**) Stretch Sensor 1, (**b**) Stretch Sensor 2, and (**c**) Stretch Sensor 3, used for the smart brace according to the elongation expressed as strain. The initial length of the stretch sensors (strain equal to 0%) is 100 mm. The red lines are the lines of best fit for each set of measured data.

**Figure 5 sensors-22-05815-f005:**
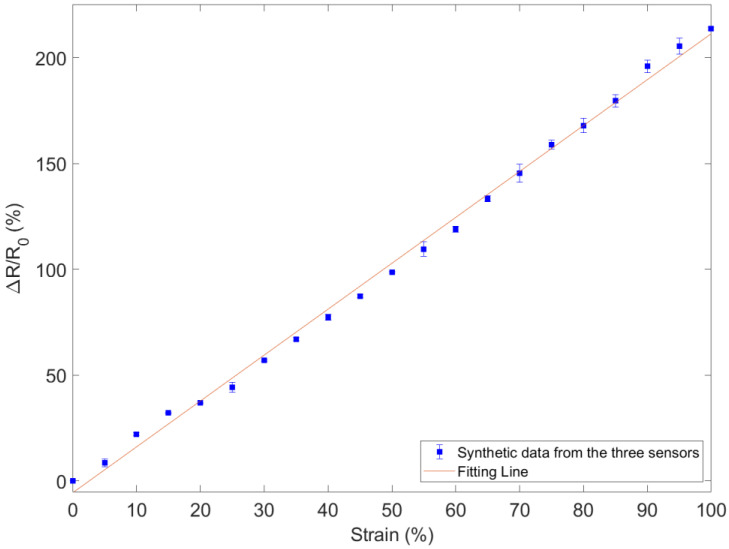
Average response of the three stretch sensors according to the elongation. The points are the average values of the three sensors (see Figure 4) and the error bars are the response variability of the three sensors. The line is the line of best fit for the measurement points.

**Figure 6 sensors-22-05815-f006:**
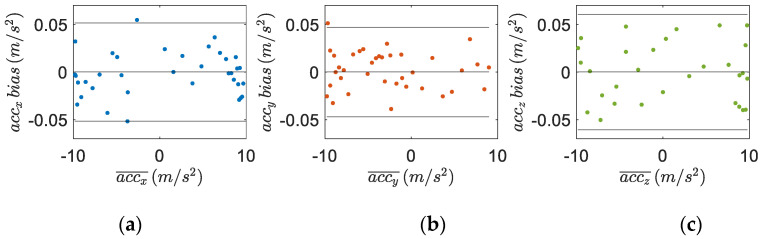
Bland–Altman plots to assess the agreement of the tibial IMU with the commercial, gold standard, Xsens system on the (**a**) x-axis, (**b**) y-axis and (**c**) z-axis of the two accelerometers. The horizontal axis is the average acceleration of the two instruments (i.e., accelerometer units), while the vertical axis is the difference between the two. The solid horizontal lines are the average difference and the limits of agreement.

**Figure 7 sensors-22-05815-f007:**
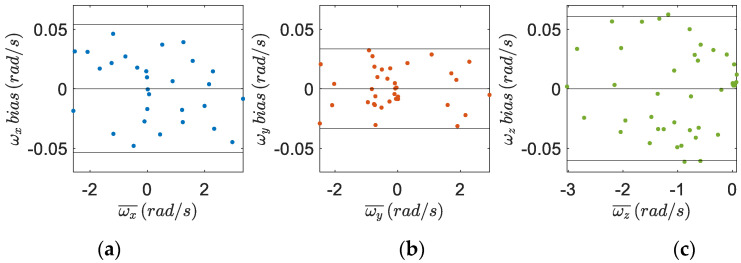
Bland–Altman plots to assess the agreement of the tibial IMU with the commercial, gold standard, Xsens system on the (**a**) x-axis, (**b**) y-axis and (**c**) z-axis of the two gyroscopes. The horizontal axis is the average angular velocity of the two instruments (gyroscopes units), while the vertical axis is the difference between the two. The solid horizontal lines are the average difference and the limits of agreement.

**Figure 8 sensors-22-05815-f008:**
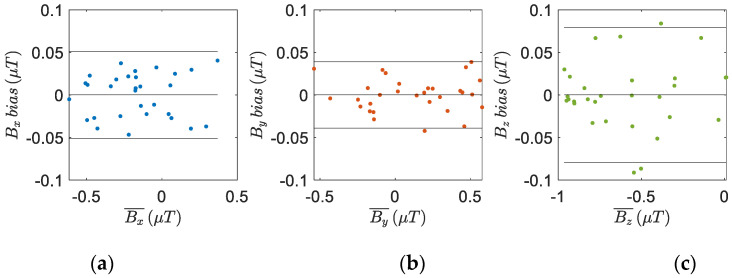
Bland–Altman plots to assess the agreement of the tibial IMU with the commercial, gold standard, Xsens system on the (**a**) x-axis, (**b**) y-axis and (**c**) z-axis of the two magnetometers. The horizontal axis is the average magnetic field strength of the two instruments (magnetometer units), while the vertical axis is the difference between the two. The solid horizontal lines are the average difference and the limits of agreement.

**Figure 9 sensors-22-05815-f009:**
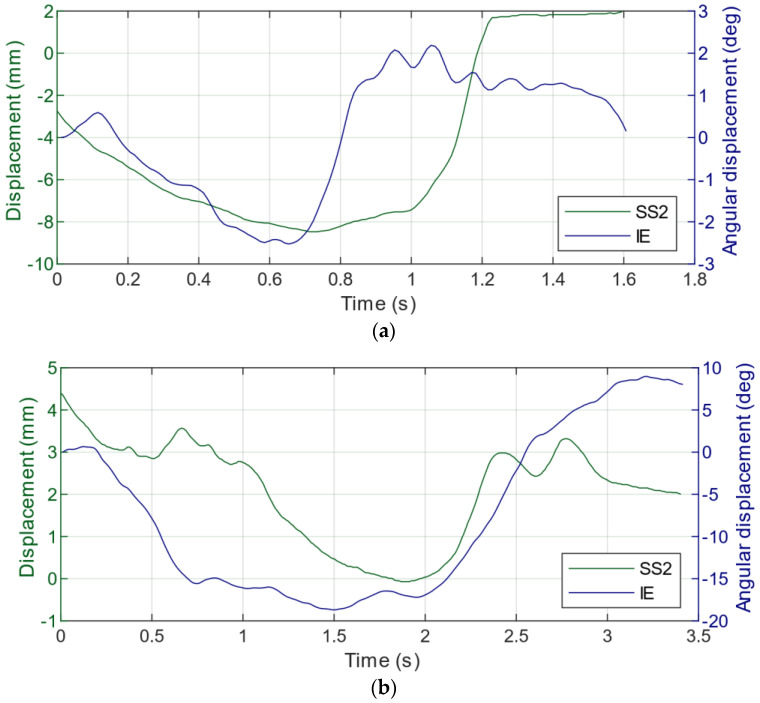
Stretch Sensor 2 displacement (green) and IE angular displacement (blue) collected through the smart brace from healthy (**a**) and injured ACL (**b**) during pivot shift test.

**Figure 10 sensors-22-05815-f010:**
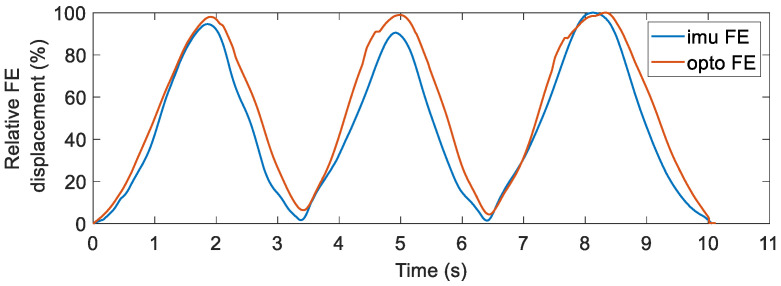
Relative flexion and extension displacement (FE) of the knee during the passive flexion and extension test calculated from measurements of the optoelectronic system (*opto*) and those of the IMUs of the smart brace (*imu*).

**Figure 11 sensors-22-05815-f011:**
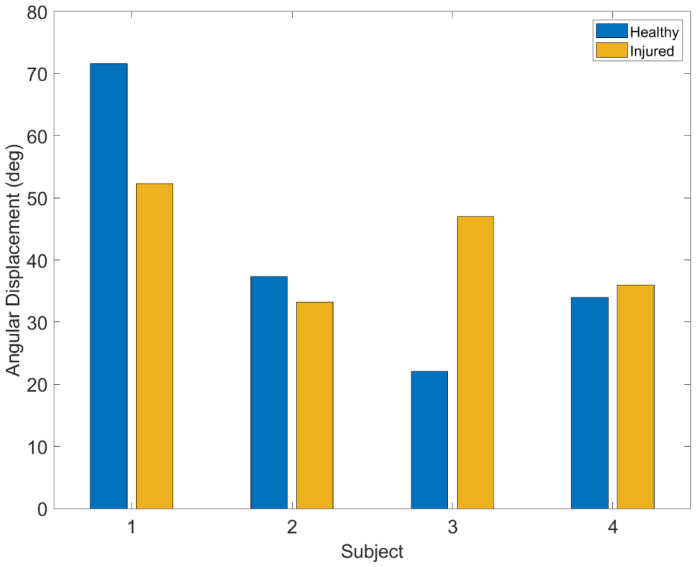
Angular displacement of the IE of healthy and the injured knees for each tested subject during the PS test.

**Figure 12 sensors-22-05815-f012:**
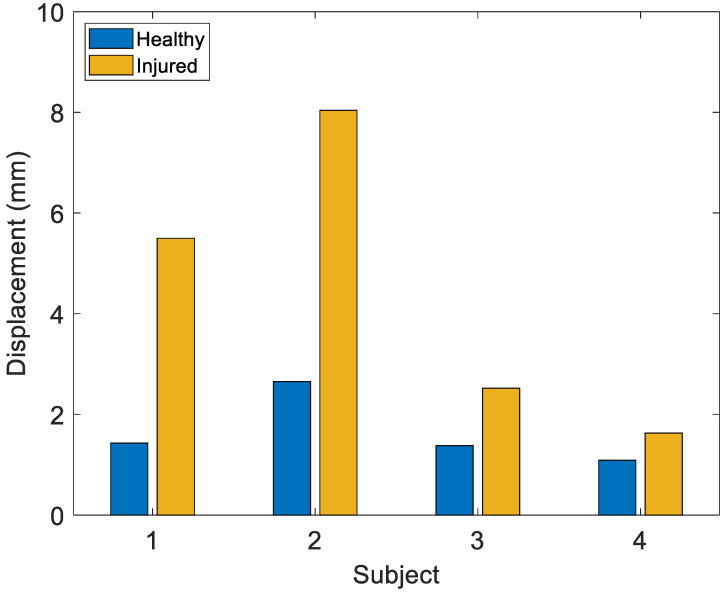
Maximum displacement for Stretch Sensor 3 in the Drawer test for the healthy and injured knees in four subjects.

**Figure 13 sensors-22-05815-f013:**
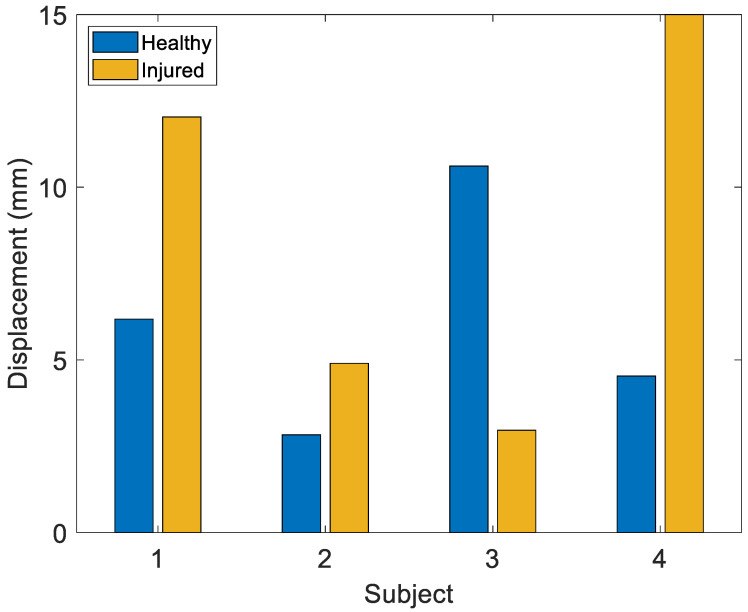
Maximum displacement for Stretch Sensor 3 in the Lachman test for the healthy and injured knees in four subjects.

**Figure 14 sensors-22-05815-f014:**
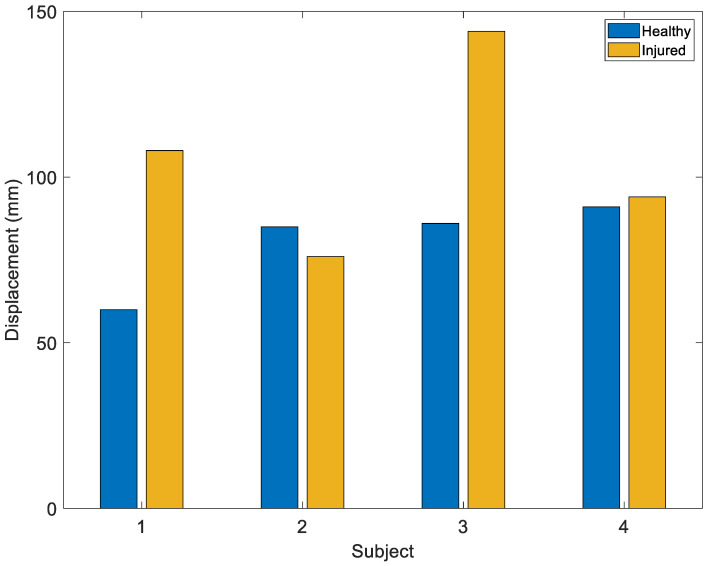
Maximum strain of Stretch Sensor 2 during pivot shift test.

**Table 1 sensors-22-05815-t001:** Results of the validation tests for the IMUs of the smart brace. FE is the knee flexion and extension, AA is the adduction and abduction, and IE is the internal and external rotation.

Type of Test	Parameter	Angle		
		FE	AA	IE
Passive Flexion and Extension	*Span_opto*	85°	12°	19°
	*Span_imu*	74°	18°	20°
	*E_max*	30°	6°	7°
	*E_point*	23°	5°	9°
	*E_scal*	12%	25%	32%
Pivot Shift Test	*Span_opto*	78°	13°	24°
	*Span_imu*	67°	17°	45°
	*E_max*	24°	4°	13°
	*E_point*	14°	5°	21°
	*E_scal*	14%	22%	23%
Lachman Test	*Span_opto*	12°	10°	14°
	*Span_imu*	13°	7°	11°
	*E_max*	2°	2°	2°
	*E_point*	2°	2°	4°
	*E_scal*	16%	15%	36%
Drawer Test	*Span_opto*	5°	5°	11°
	*Span_imu*	4°	5°	7°
	*E_max*	1°	2°	3°
	*E_point*	1°	1°	3°
	*E_scal*	23%	25%	32%

## Data Availability

Not applicable.

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
