# Peer review of "Smart Brace for Static and Dynamic Knee Laxity Measurement"

_sensors, 2022, doi:10.3390/s22155815_

Round 1
Reviewer 1 Report
The manuscript discussed the development and testing of a new brace device for assessment of either static or dynamic laxity. The system includes IMU and stretchable sensors.
Abstract: The abstract will be greatly improved if quantitative results are added and also mentioning how this will benefit further applications.
P3L111. In this paragraph a more extended introduction of the topic should be added, i.e. what methods, sensors, etc. where used and why.
P6L199 : authors need to specify the number of hours to be more clear on the description, if only 45mA are considered, this would be around 22 hours, but need to consider all the sensors.
Section 3 should also include a characterisation of the stretchable sensors.
what are the reaction and recovery times?
How were the sensors calibrated?
What was the procedure to take the measurements?
Did you consider the overshoot and stabilisation time of the
stretchable sensors and how this affected the measurements?
How are the IMU and stretchable sensors used in combination, Will be necessary the use of stretchable sensors or only the use of IMU sensors will be enough for the smart brace?
Table 1 discusses the results between opto and IMU, why the stretchable sensors are not used in this comparison?
The results are mainly focused in sensors 3, but does not include results and discussions of the other two sensors.
in the pivot test, the 3 sensors need to be considered to determine the flexion as well as the lateral displacement.
Reviewer 2 Report
The paper “Smart Brace for Static and Dynamic Knee Laxity Measurement” study focuses on the long-standing dream of instrumental objectification of clinical testing for ACL rupture.
This work has a significant technical part in the development of the Smart knee brace and its subsequent validation.
It is possible that the article in this form is heavily overloaded and could be divided into two parts: the development of the Smart knee brace and its validation. And the second article is already a study of healthy and people with ACL damage. Each article has its own reader and his education and experience differ significantly. For the first one, these are engineering personnel, for the second, orthopedic doctors. Sensors magazine is perhaps one of the few where an article with such a diverse audience of readers will be relevant. Therefore, I do not insist on such a division.
As a matter of fact, I have the following comments.
Line 162 - Why did you choose to locate the sensor on the soft tissues and not on the facies lateralis of tibia?
Figures 6, 7 and 8. In the caption of this figure, you need to add what each of them means (letters - A, B, C). Of course, it is clear that these are three planes X,Y,Z, but which one? In addition, figure 7 contains double indexing (a), (b), (c).
The results did not mention Stretch Sensor 1 data. Apparently, they are not indicative, but I think that a few words should be said about this.
I think it would be interesting for readers to see if you would give any test, for example, a graph of resistance changes for all three Stretch Sensors, of course graphs for a healthy and injured knee joint. In continuation of this, it may also be useful for the reader to see a graph for any of the tests (and maybe all) where IMU and Stretch Sensors data will be presented. The dimension there is very different, so that it can rather be like two graphs for one of the tests. It seems to me that this is a clear way to show the potential of the Smart knee brace as a possible way to objectify the clinical testing of an ACL tear.
In the Conclusion section, it seems necessary to comment on how you see the information for the doctor that he can get from the Smart knee brace, conditionally, when the device goes beyond the model? Again about Stretch Sensor 1 - its data was not useful and this will simplify the device?
